# Body mass index and severity/fatality from coronavirus disease 2019: A nationwide epidemiological study in Korea

In Sook Kang[1]*, Kyoung Ae Kong[2]

1 Division of Cardiology, Department of Internal Medicine, Ewha Womans University Mokdong Hospital, College of Medicine, Ewha Womans University, Seoul, Republic of Korea, 2 Department of Preventive Medicine, College of Medicine, Ewha Womans University, Seoul, Republic of Korea

* pinkvision@ewha.ac.kr

**Data Availability Statement:** Data cannot be shared publicly for protecting personal information by the Korea Disease Control and Prevention Agency (KDCA). Broad information regarding Korean COVID-19 statistics are released daily on a

## Abstract

### Objective

Obesity has been reported as a risk factor for severe coronavirus disease 2019 (COVID-19) in recent studies. However, the relationship between body mass index (BMI) and COVID-19 severity and fatality are unclear.

### Research design and methods

This study included 4,141 COVID-19 patients who were released from isolation or had died as of April 30, 2020. This nationwide data was provided by the Korean Centers for Disease Control and Prevention Agency. BMI was categorized as follows; < 18.5 kg/m$^2$, 18.5–22.9 kg/m$^2$, 23.0–24.9 kg/m$^2$, 25.0–29.9 kg/m$^2$, and $\geq$ 30 kg/m$^2$. We defined a fatal illness if the patient had died.

### Results

Among participants, those with a BMI of 18.5–22.9 kg/m$^2$ were the most common (42.0%), followed by 25.0–29.9 kg/m$^2$ (24.4%), 23.0–24.9 kg/m$^2$ (24.3%), $\geq$ 30 kg/m$^2$ (4.7%), and < 18.5 kg/m$^2$ (4.6%). In addition, 1,654 (41.2%) were men and 3.04% were fatalities. Multi-variable analysis showed that age, male sex, BMI < 18.5 kg/m$^2$, BMI $\geq$ 25 kg/m$^2$, diabetes mellitus, chronic kidney disease, cancer, and dementia were independent risk factors for fatal illness. In particular, BMI < 18.5 kg/m$^2$ (odds ratio [OR] 3.97, 95% CI 1.77–8.92), 25.0–29.9 kg/m$^2$ (2.43, 1.32–4.47), and $\geq$ 30 kg/m$^2$ (4.32, 1.37–13.61) were found to have higher ORs than the BMI of 23.0–24.9 kg/m$^2$ (reference). There was no significant difference between those with a BMI of 18.5–22.9 kg/m$^2$ (1.59, 0.88–2.89) and 23.0–24.9 kg/m$^2$.

### Conclusions

This study demonstrated a non-linear (U-shaped) relationship between BMI and fatal illness. Subjects with a BMI of < 18.5 kg/m$^2$ and those with a BMI $\geq$ 25 kg/m$^2$ had a high risk of fatal

public web site (http://ncov.mohw.go.kr/en/) and through the media by KCDA. You may contact KCDA for the detailed data in current study.

**Funding:** The authors received no specific funding for this work.

**Competing interests:** The authors have declared that no competing interests exist.

illness. Maintaining a healthy weight is important not only to prevent chronic cardiometabolic diseases, but also to improve the outcome of COVID-19.

## Introduction

Since the first coronavirus disease 2019 (COVID-19) case was reported in December 2019 in China, the pandemic has been progressing worldwide. Nowadays, people are more likely to gain weight as a result of increasing social distancing, increasing intake of unhealthy food, and decreased physical activity [1]. Obesity and overweight are related to metabolic syndrome and cardiovascular events, and are important health issues that can cause various chronic illnesses [2]. Previous studies of body mass index (BMI) showed that mortality due to cardiovascular diseases and all other causes are increased in both underweight and obesity, which can be represented by a U-shaped curve [3–6].

Importantly, obesity adversely affects the immune system, and consequently increases the risk of infection [2]. Furthermore, obesity is a risk factor for severe COVID-19 [7]. In a French study, patients with BMI > 35 kg/m$^2$ had a higher odds ratio of 7.36, than those with BMI < 25 kg/m$^2$ for invasive mechanical ventilation among patients admitted to the intensive care unit (ICU) due to COVID-19 [8]. From a study in China, obese men (BMI $\geq$ 28 kg/m$^2$) were associated with increased risk of severe COVID-19 than normal weight men (BMI < 24 kg/m$^2$) with an odds ratio of 5.66 (95% CI; 1.8–17.75) [9]. Recent studies using Mendelian randomization showed a higher genetically proxied BMI-related increasing risk of severe respiratory COVID-19 and COVID-19 hospitalization [10, 11].

Previously, cardiac disease and obesity have also been reported as common comorbidities in the Middle East respiratory syndrome [12]. Similarly, in the 2009 influenza pandemic, obesity was reported as a common comorbidity in critically ill patients admitted to the ICU [13]. However, one study showed an apparent decrease in the rate of pneumonia with increasing BMI, and pneumonia was more common in underweight individuals during the same influenza pandemic [14]. Furthermore, the obesity survival paradox (the inverse relationship between obesity and mortality) has also been reported in patients with pneumonia in a meta-analysis [15]. Hence, this issue remained controversial, as other studies have showed that there is no association between BMI and survival during the influenza pandemic [16].

To date, there is scarcity of information regarding the obesity survival paradox or underweight risk associated with the outcomes of COVID-19. A systematic review found that the association of obesity with poor clinical presentation and the need for hospitalization due to COVID-19 was consistent, but the association of obesity mortality was not [17]. Hence, in this study, we aimed at evaluating the relationship between BMI and severe COVID-19, especially fatality.

## Materials and methods

### Data source and study population

This study was performed using nationwide data of COVID-19 confirmed patients who were released from isolation or who died from January 19 to April 30, 2020, in the Republic of Korea. The Korean government has centralized control systems for medical insurance and disease control. The Korea Disease Control and Prevention Agency (KDCA) has been actively tracking and managing almost all confirmed cases and contactors from the beginning of the COVID-19 outbreak. They constructed a nationwide registry of COVID-19 and provided the data (anonymized without personal information) to researchers with granted permission. We

temporarily accessed the data from the KDCA through an encrypted remote system that contained anonymized data of 5,628 patients who were released from isolation or died during the aforementioned period. After excluding 1,487 patients (272 patients under the age of 20, 19 pregnant women, 26 without clinical severity information, 4 with no information regarding comorbidity, and 1,166 without available BMI value), 4,141 patients were included in the analysis for this study.

The Ewha Womans University Mokdong Hospital Institutional Review Board deemed this study exempt from ethical review and waived the requirement for informed consent (EUMC2020-07-002), since it was completely anonymized and without personal information.

## Study definitions and outcome assessment

COVID-19 was diagnosed based on the detection of severe acute respiratory syndrome coronavirus 2 (SARS-CoV-2) by real-time polymerase chain reaction test with respiratory specimens [18]. The investigation of comorbid diseases was performed depending on whether the patients were previously diagnosed with specific diseases or not. Body temperature and BMI were the initial findings of hospital admission.

BMI was calculated as body weight in kilograms divided by height in meter squared and categorized as followings [19–21]: $< 18.5$ kg/m$^2$, 18.5–22.9 kg/m$^2$, 23.0–24.9 kg/m$^2$, 25.0–29.9 kg/m$^2$, and $\geq 30$ kg/m$^2$.

Disease severity was defined according to the guidelines of the KDCA and World Health Organization (WHO) [22, 23]. Briefly, we defined a patient as having a "critical illness" if they required more than invasive mechanical ventilation and "fatal illness" if they died. Critical illnesses included patients requiring invasive mechanical ventilation, those with multi-organ failure, those requiring extracorporeal membrane oxygenation therapy, and/or those who died. The severity evaluation was based on the most severe condition during the hospital stay. Critical illness was defined as severe COVID-19.

## Statistical analysis

Descriptive statistics were described using frequencies and proportions for categorical variables. The chi-square test was used to compare the variables. The values of continuous variables were expressed as mean and standard deviation. Analysis of variance was performed for the body temperature.

Multivariable logistic regression was used to analyze the independent risk for COVID-19 critical illness and COVID-19 fatal illness after adjusting for age, sex, BMI, and five comorbid diseases: diabetes mellitus (DM), hypertension, chronic kidney disease (CKD), cancer (excluding cured cases) and dementia. The age variable was given as a categorical variable in units of 10 years, but we calculated the odds ratio of every 10 years as a continuous variable. Since dementia had missing data (n = 314) that were not replaced, 3827 participants were included in the multivariable analysis. Although the BMI of 23.0–24.9 kg/m$^2$ was classified as overweight rather than normal-weight according to the Asia-Pacific cutoff values [21], we used it as a reference. Since this group is of normal weight according to the WHO classification [24], and a previous study reported a BMI cutoff value of Koreans as 24.2 kg/m$^2$ [25].

To analyze the relationship between BMI and fatality by sex, we performed a multivariable analysis after adjusting for age and the five comorbid conditions. In addition to sex, we analyzed the relationship between BMI and fatality according to whether they had DM or HTN. We performed multivariable analysis after adjusting for age, sex, and four comorbidities other than the target disease (DM or HTN). Then, the relationships between BMI and fatality in the groups with and without the diseases (DM or HTN) were evaluated separately.

The threshold for statistical significance was set at $P < 0.05$. Statistical analyses were performed using SAS software (version 9.4, SAS Institute, Cary, NC, USA).

## Results

Baseline characteristics according to BMI are presented in Table 1. A BMI of 18.5–22.9 kg/m$^2$ was the most common, with 1,741 (42.0%) patients, followed by BMI of 25.0–29.9, 23.0–24.9 kg/m$^2$ as 1,011 (24.4%), and 1,005 (24.3%), respectively. Moreover, BMI of < 18.5, and ≥ 30 kg/m$^2$ were less than 5%. In the BMI < 18.5, 18.5–22.9, and ≥ 30 kg/m$^2$ groups, the most patients were in their 20s, whereas in the BMI of 23.0–24.9, and 25.0–29.9 kg/m$^2$ groups, most patients were in their 50s. The BMI of 18.5–22.9 kg/m$^2$ was the most common both in male and female, whereas the proportions of the BMI ranging 23.0–24.9 and 25.0–29.9 kg/m$^2$ were higher in male than in female (raw %, male: 27.3, 32.0 vs. female: 22.1, 19.0, respectively). Those with BMI ≥ 25.0 kg/m$^2$ had higher body temperature than others ($P < 0.0001$). Dyspnea was a more frequent complaint in patients with a BMI of < 18.0 and ≥ 25.0 kg/m$^2$ than in others ($P = 0.0135$). Hypertension and DM were more common in the BMI of 23.0–24.9 and ≥ 25.0 kg/m$^2$ than in others ($P < 0.0001$).

S1 Table shows the patient characteristics according to the fatal illness. We noted that 41.7% were male. There were no deaths in patients aged < 40 years. The overall proportions of patients in their 20s and 50s were 942 (22.7%) and 873 (21.1%), respectively. Systolic blood pressures of < 120 mmHg and ≥ 140 mmHg were more related to fatal illness than those between 120 mmHg and 140 mmHg ($P < 0.0001$).

Independent risk factors associated with severe COVID-19 are shown in Table 2. With every 10 years of increase in age, the odds ratio of multivariable analyses for critical and fatal illness increased to 2.64 (95% CI: 2.20–3.15) and 3.04 (2.45–3.78), respectively. Male sex was also found to be an independent risk factor. When the BMI of 23.0–24.9 kg/m$^2$ group was used as a reference, the risk of BMI ranging 18.5–22.9 kg/m$^2$ was as high as 1.59 (0.88–2.89) for fatal illness without statistical significance. All five comorbidities (DM, hypertension, CKD, cancer, and dementia) were related to a higher critical illness rate compared to the absence of comorbidities. Hypertension had an odds ratio of 1.34 in fatal illness, but had no statistical power (95% CI: 0.96–2.30) in multivariable analysis.

S1 Fig shows the number of critical and fatal illnesses according to the BMI group and the fatality rate of the study population. The BMI of 23.0–24.9 kg/m$^2$ group was found to have the lowest fatality. Fatality rate increased in both BMI of < 18.5 and 25.0–29.9 kg/m$^2$ groups, but slightly decreased in the BMI of ≥ 30.0 kg/m$^2$ group. As shown in S1 Fig, patient fatality was the highest in the BMI of < 18.5 kg/m$^2$, followed by the BMI of 25.0–29.9, and ≥ 30.0 kg/m$^2$. After multivariable analysis with adjusting covariates, a similar result was shown in the group with BMI < 18.5 kg/m$^2$ (Fig 1). However, the result according to the degree of obesity showed that the odds ratio of BMI ≥ 30.0 kg/m$^2$ was higher in patients with critical and fatal illness than in those with a BMI of 25.0–29.9 kg/m$^2$. Hence, the risk was increased in the BMI < 18.5 kg/m$^2$ and BMI of ≥ 25.0 kg/m$^2$ groups, resulting in a U-shaped curve for critical and fatal illness (Fig 1).

Fig 2 shows the association between BMI and sex for critical and fatal illnesses. This was expressed as an odds ratio after adjusting for age and comorbidities. Fig 2 shows similar U-shaped curves in both males and females, although it seems to be slightly different in detail depending on sex. Women showed a higher risk in their BMI < 18.5 kg/m$^2$ population than men. While, men showed a higher risk in their BMI ≥ 25.0 kg/m$^2$ population than women (Fig 2A and 2B). However, the $P_{interaction}$ for among males and females were not significant.

Similar non-linear relationships are shown in Fig 3. This figure shows subgroup analyses of subjects with (A) DM or (B) hypertension. Each group showed a non-linear pattern with

**Table 1. Baseline characteristics of the study population according to the body mass index.**

| | Body mass index (kg/m$^2$), N (column %) | | | | | |
| --- | --- | --- | --- | --- | --- | --- |
| | <18.5 | 18.5–22.9 | 23.0–24.9 | 25.0–29.9 | ≥30 | p |
| N (row %) | 191 (4.6) | 1741 (42.0) | 1005 (24.3) | 1011 (24.4) | 193 (4.7) | |
| Age (years) | | | | | | <0.0001 |
| 20–29 | 68 (35.6) | 455 (26.1) | 172 (17.1) | 193 (19.1) | 54 (28.0) | |
| 30–39 | 24 (12.6) | 183 (10.5) | 102 (10.1) | 117 (11.6) | 43 (22.3) | |
| 40–49 | 13 (6.8) | 270 (15.5) | 133 (13.2) | 145 (14.3) | 34 (17.6) | |
| 50–59 | 20 (10.5) | 347 (19.9) | 241 (24.0) | 234 (23.1) | 31 (16.1) | |
| 60–69 | 16 (8.4) | 249 (14.3) | 204 (20.3) | 186 (18.4) | 19 (9.8) | |
| 70–79 | 28 (14.7) | 136 (7.8) | 111 (11.0) | 95 (9.4) | 8 (4.1) | |
| ≥ 80 | 22 (11.5) | 101 (5.8) | 42 (4.2) | 41 (4.1) | 4 (2.1) | |
| Female | 135 (70.7) | 1182 (67.9) | 534 (53.1) | 459 (45.4) | 105 (54.4) | <0.0001 |
| Male | 56 (29.3) | 559 (32.1) | 471 (46.9) | 552 (54.6) | 88 (45.6) | |
| BT (°C) | 36.97 ± 0.5 | 36.91 ± 0.5 | 36.92 ± 0.5 | 37.02 ± 0.6 | 37.05 ± 0.6 | <0.0001 |
| Dyspnea | | | | | | 0.0135 |
| Yes | 26 (13.6) | 176 (10.1) | 118 (11.7) | 146 (14.4) | 26 (13.5) | |
| No | 165 (86.4) | 1565 (89.9) | 886 (88.2) | 865 (85.6) | 167 (86.5) | |
| SBP* (mmHg) | | | | | | <0.0001 |
| <120 | 81 (42.4) | 529 (30.4) | 183 (18.2) | 152 (15.0) | 19 (9.8) | |
| 120–129 | 36 (18.8) | 387 (22.2) | 219 (21.8) | 207 (20.5) | 31 (16.1) | |
| 130–139 | 27 (14.1) | 344 (19.8) | 205 (20.4) | 209 (20.7) | 55 (28.5) | |
| 140–159 | 30 (15.7) | 363 (20.9) | 291 (29.0) | 329 (32.5) | 65 (33.7) | |
| ≥ 160 | 16 (8.4) | 115 (6.6) | 107 (10.6) | 113 (11.2) | 23 (11.9) | |
| Diabetes mellitus | | | | | | <0.0001 |
| Yes | 18 (9.4) | 157 (9.0) | 137 (13.6) | 160 (15.8) | 29 (15.0) | |
| No | 173 (90.6) | 1584 (91.0) | 868 (86.4) | 851 (84.2) | 164 (85.0) | |
| Hypertension | | | | | | <0.0001 |
| Yes | 28 (14.7) | 234 (13.4) | 247 (24.6) | 301 (29.8) | 52 (26.9) | |
| No | 163 (85.3) | 1507 (86.6) | 758 (75.4) | 710 (70.2) | 141 (73.1) | |
| Heart failure | | | | | | 0.0132 |
| Yes | 6 (3.1) | 12 (0.7) | 8 (0.8) | 13 (1.3) | 1 (0.5) | |
| No | 185 (96.9) | 1729 (99.3) | 997 (99.2) | 998 (98.7) | 192 (99.5) | |
| Chronic heart disease** | | | | | | 0.0686 |
| Yes | 7 (3.7) | 42 (2.4) | 40 (4.0) | 41 (4.1) | 4 (2.1) | |
| No | 183 (95.8) | 1693 (97.2) | 961 (95.6) | 965 (95.5) | 189 (97.9) | |
| Asthma | | | | | | 0.0333 |
| Yes | 3 (1.6) | 31 (1.8) | 32 (3.2) | 23 (2.3) | 9 (4.7) | |
| No | 188 (98.4) | 1710 (98.2) | 973 (96.8) | 988 (97.7) | 184 (95.3) | |
| Chronic obstructive lung disease | | | | | | 0.0006 |
| Yes | 6 (3.1) | 14 (0.8) | 5 (0.5) | 3 (0.3) | 1 (0.5) | |
| No | 185 (96.9) | 1727 (99.2) | 1000 (99.5) | 1008 (99.7) | 192 (99.5) | |
| Chronic kidney disease | | | | | | 0.7807 |
| Yes | 3 (1.6) | 18 (1.0) | 8 (0.8) | 12 (1.2) | 3 (1.6) | |
| No | 188 (88.4) | 1723 (99.0) | 997 (99.2) | 999 (98.8) | 190 (98.4) | |
| Malignancy | | | | | | 0.2349 |
| Yes | 7 (3.7) | 48 (2.8) | 26 (2.6) | 18 (1.8) | 8 (4.1) | |
| No | 184 (98.4) | 1693 (97.2) | 979 (99.2) | 993 (98.2) | 185 (95.9) | |
| Dementia¶ | | | | | | <0.0001 |

(*Continued*)

**Table 1.** (Continued)

| | Body mass index (kg/m$^2$), N (column %) | | | | | |
| | <18.5 | 18.5–22.9 | 23.0–24.9 | 25.0–29.9 | ≥30 | p |
|---|---|---|---|---|---|---|
| Yes | 20 (10.5) | 65 (3.7) | 20 (2.0) | 16 (1.6) | 0 | |
| No | 162 (84.8) | 1528 (87.8) | 911 (90.6) | 924 (91.4) | 181 (93.8) | |
| **Disease severity** | | | | | | 0.0003 |
| Non critical | 175 (91.6) | 1686 (96.8) | 980 (97.5) | 962 (95.2) | 186 (96.4) | |
| Critical | 16 (8.4) | 55 (3.2) | 25 (2.5) | 49 (4.8) | 7 (3.6) | |
| **Discharge** | | | | | | <0.0001 |
| Live | 175 (91.6) | 1695 (97.4) | 985 (98.0) | 972 (96.1) | 188 (97.4) | |
| Death | 16 (8.4) | 46 (2.6) | 20 (2.0) | 39 (3.9) | 5 (2.6) | |

BT, body temperature, mean ± standard deviation; SBP, systolic blood pressure; DM, diabetes mellitus; CHD, chronic heart disease; COPD, chronic obstructive pulmonary disease; CKD, chronic kidney disease.

Missing data

* n = 5

** n = 16

¶ n = 314.

increasing BMI. In addition, there was no significant difference according to the presence or absence of the diseases ($P_{interaction}$ = 0.62 for DM, $P_{interaction}$ = 0.19 for hypertension)

## Discussion

In this nationwide study, we showed that BMI ≥ 25.0 kg/m$^2$ was associated with an increased risk for fatal illness as well as critical illness from COVID-19. Furthermore, BMI < 18.5 kg/m$^2$

**Table 2. Logistic analyses of in critical and fatal illness patients.**

| | Critical illness | | Fatal illness | |
| | Odds ratio (95% CI) | | Odds ratio (95% CI) | |
| | Univariable | Multivariable* | Univariable | Multivariable* |
|---|---|---|---|---|
| **Age**** | 3.09 (2.65–3.60) | 2.64 (2.20–3.15) | 3.65 (3.03–4.39) | 3.04 (2.45–3.78) |
| **Male (vs. female)** | 1.97 (1.42–2.74) | 2.85 (1.94–4.19) | 1.90 (1.33–2.72) | 2.92 (1.90–4.48) |
| **BMI (kg/m$^2$)** | | | | |
| <18.5 | 3.58 (1.88–6.85) | 3.29 (1.54–7.04) | 4.50 (2.29–8.86) | 3.97 (1.77–8.92) |
| 18.5–22.9 | 1.28 (0.79–2.07) | 1.59 (0.94–2.71) | 1.34 (0.79–2.27) | 1.59 (0.88–2.89) |
| 23–24.9 | 1 (Reference) | 1 (Reference) | 1 (Reference) | 1 (Reference) |
| 25.0–29.9 | 2.00 (1.22–3.26) | 2.37 (1.38–4.07) | 1.98 (1.14–3.41) | 2.43 (1.32–4.47) |
| ≥30 | 1.48 (0.63–3.46) | 4.40 (1.66–11.66) | 1.31 (0.49–3.53) | 4.32 (1.37–13.61) |
| **Diabetes mellitus** | 5.74 (4.10–8.04) | 1.95 (1.32–2.86) | 6.42 (4.46–9.25) | 2.38 (1.50–3.51) |
| **Hypertension** | 6.60 (4.72–9.23) | 1.64 (1.11–2.43) | 6.47 (4.48–9.33) | 1.48 (0.96–2.30) |
| **CKD** | 10.6 (5.35–21.02) | 3.12 (1.36–7.19) | 10.1 (4.87–20.92) | 2.74 (1.11–6.76) |
| **Cancer** | 3.88 (2.12–7.09) | 2.28 (1.12–4.62) | 4.80 (2.61–8.83) | 3.01 (1.44–6.28) |
| **Dementia** | 12.48 (8.09–19.26) | 2.13 (1.25–3.63) | 16.17 (10.36–25.22) | 2.50 (1.43–4.36) |

*n = 3827, The ORs of multivariable analyses were adjusted for age, sex, and five comorbidities (diabetes mellitus, hypertension, chronic kidney disease, cancer, and dementia).

**Age was given as a categorical variable in units of 10 years, but we calculated the odds ratio of every 10 years as a continuous variable. BMI, body mass index; CKD, chronic kidney disease.

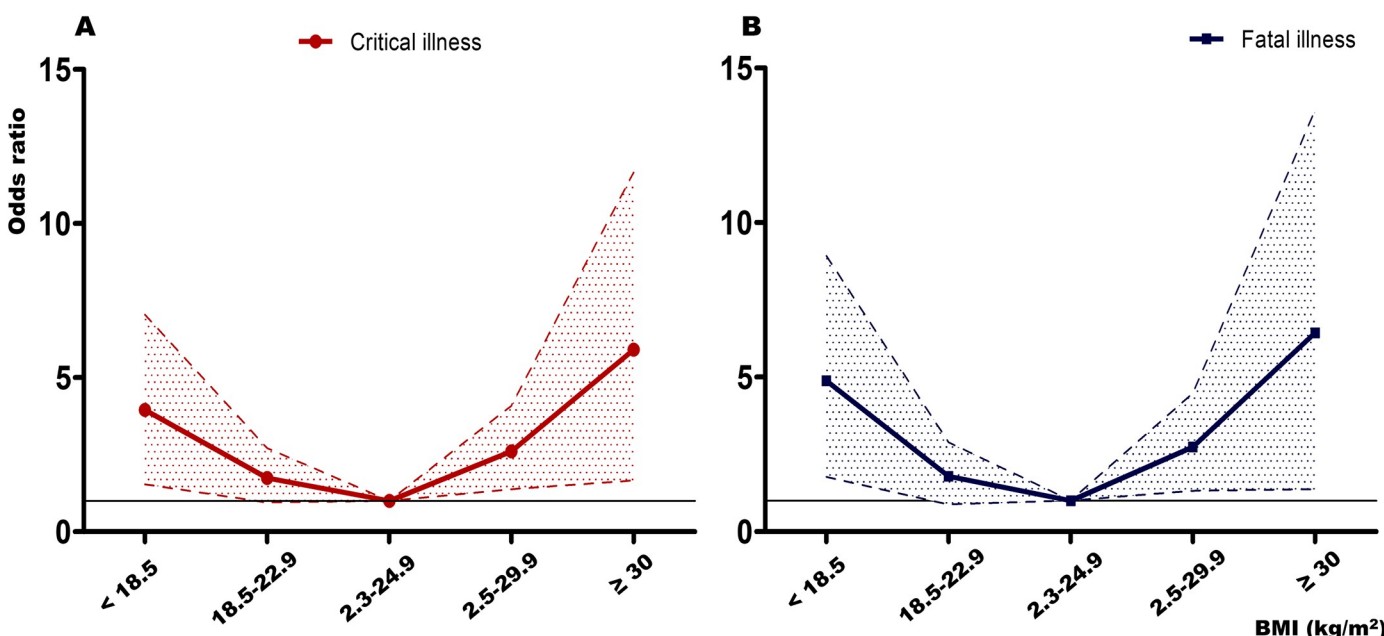

**Fig 1. Odds ratios for critical and fatal illness according to body mass index.** Odds ratios were adjusted for age, gender, and five comorbidities (diabetes mellitus, hypertension, chronic kidney disease, cancer, and dementia).

also increased the risk of critical and fatal illness caused by COVID-19, similar to that seen with BMI $\geq$ 25.0 kg/m$^2$. The frequency data of S1 Fig shows that fatality decreased in those with BMI $\geq$ 30.0 kg/m$^2$, and it appeared as if the obesity paradox existed. However, the odds ratio after adjusting for covariates showed a U-shaped curve, as shown in Fig 1. Hence, BMI $\geq$ 25.0 kg/m$^2$ and < 18.5 kg/m$^2$ were found to have an association with increased fatality from COVID-19 independent of combined comorbidities. Although the male sex showed a higher risk of fatal illness from COVID-19, there was no definite evidence that BMI had a more detrimental effect on men than on women in this study.

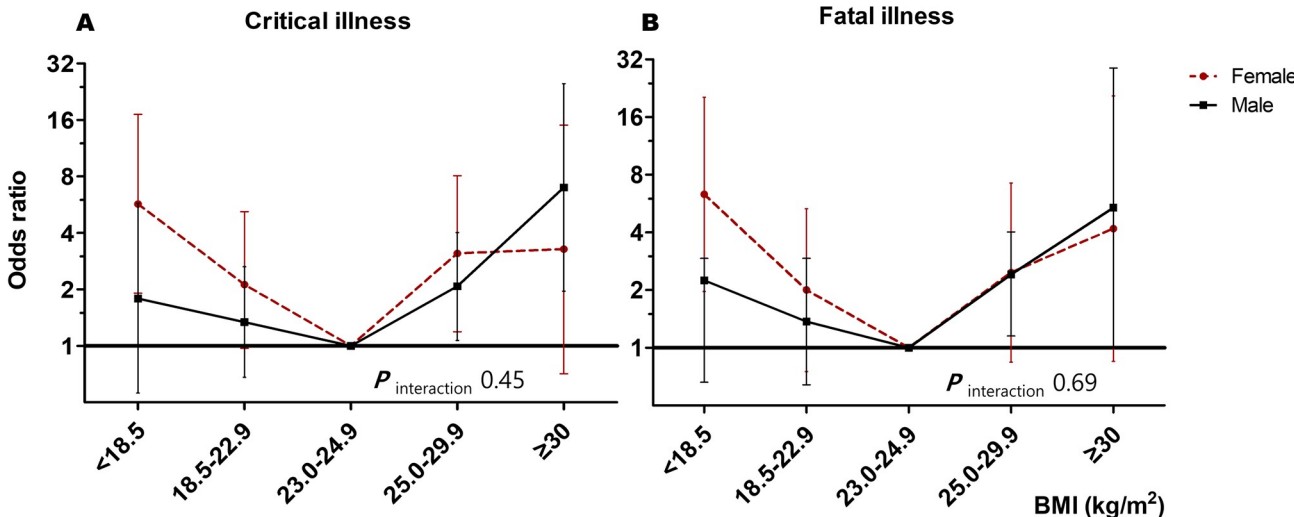

**Fig 2.** Odds ratios for critical and fatal illness of male and female according to body mass index (A) critical illness, (B) fatal illness Odds ratios were adjusted for age and five comorbidities (diabetes mellitus, hypertension, chronic kidney disease, cancer, and dementia).

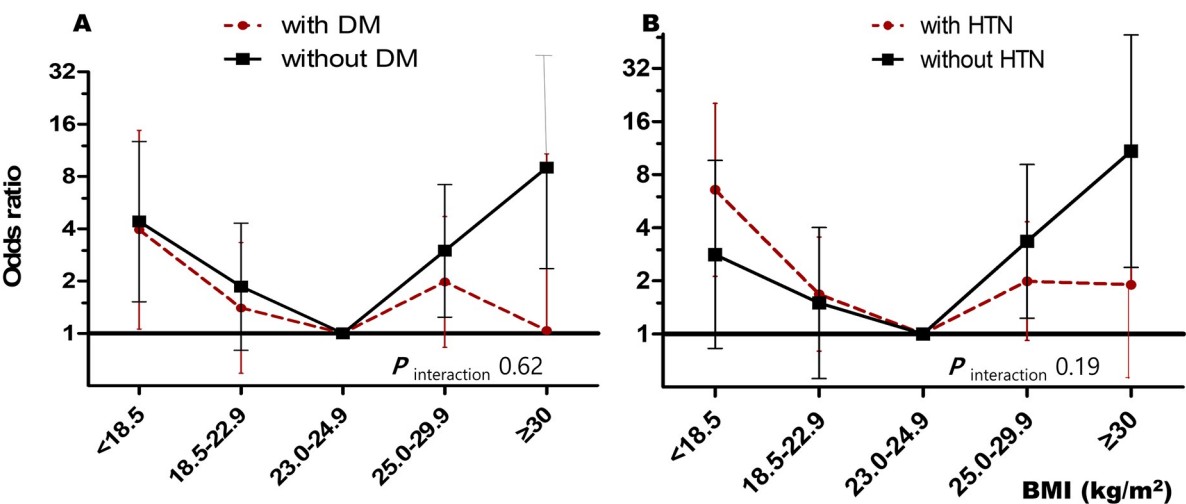

**Fig 3.** Subgroup analyses for fatal illness according to the BMI in patients with/without diabetes mellitus or hypertension (A) Diabetes Mellitus*, (B) Hypertension**. *Odds ratios were adjusted for age, gender, and four comorbidities (hypertension, chronic kidney disease, cancer, and dementia). **Odds ratios were adjusted for age, gender, and four comorbidities (diabetes mellitus, chronic kidney disease, cancer, and dementia).

The results of the current study showed that maintaining a healthy weight in the general population, regardless of underlying comorbidities, is still important in the COVID-19 pandemic era. In this study, normal weight ($18.5 < BMI < 25$ kg/m$^2$) according to the WHO criteria, especially 23–25 kg/m$^2$ in the South Korean population, is considered a healthy weight in the COVID-19 pandemic era.

Obesity, defined using BMI, has different cutoff values between the global and Asia-Pacific standards. The WHO announced the Asia-Pacific perspective: redefining obesity and its treatment' in 2000 on the evidence that the Asian population has an increased risk of diseases at BMI > 23 kg/m$^2$ [21]. The Asia-Pacific BMI was categorized as followings; underweight ($< 18.5$ kg/m$^2$), normal-weight (18.5–22.9 kg/m$^2$), overweight (23.0–24.9 kg/m$^2$), obesity-1 (25.0–29.9 kg/m$^2$), and obesity-2 ($\geq 30$ kg/m$^2$). Proposed mechanisms of this inter-population discrepancy of defined obesity by BMI include differences in percentage and distribution of body fat across populations, the interplay of genetic susceptibility, and environmental factors related to diet and sedentary lifestyle [19, 20].

However, the debate on whether racial differences exist in defining obesity based on BMI is still ongoing. Despite the differences in BMI cutoff points for obesity in these populations, a meta-analysis of four continents in 239 prospective studies showed that BMI and all-cause mortality were broadly consistent across the four continents [3]. In 2004, WHO expert consultation recommended that the use of global standards is appropriate because the BMI standards for obesity do not differ greatly by race, and it is not appropriate to stipulate different standards for the Asia-Pacific region due to small differences [19]. In a study comparing the BMI cutoff value of the South Koreans with the global standard in 2015, the cutoff point of Korean BMI was 24.2 kg/m$^2$ (sensitivity 78%, specificity 71%, as the result of receiver operating characteristic cure, based on the body fat percentage) and it was only 1.3 kg/m$^2$ lower than the global standard [25]. The result supports the recommendation of the WHO expert consultation in 2004. Nonetheless, the Korean government continues to follow the 2000 Asian-Pacific cut-off values. Hence, to avoid confusion, we did not express obesity or normal for the BMI group, but only used numeric values in this study.

There are limited studies on the clinical implications of BMI, from underweight to obesity, in viral pandemics. In the case of influenza-associated pneumonia, an increase in risk has been observed in underweight and obesity [4]. However, regarding obesity, contradictory results–obesity paradox-were reported in previous studies [15]. Similarly, there was a recent study on 'an obesity survival paradox' that authors showed the projected rates of COVID-19 infection and mortality drop with elevated prevalence of obesity in United States [26]. However, in the current COVID-19 pandemic, the majority of studies reported obesity as a risk factor for progressing to severe COVID-19 and were associated with the need for hospitalization and admission to the critical care unite due to COVID-19 [7, 17, 27, 28]. Importantly, our study also showed increased BMI as a risk factor for critical and fatal illness of COVID-19 and cannot prove the 'obesity survival paradox.'

In contrast to obesity, the relationship between underweight and COVID-19 is unknown. A recent study showed a J-shaped (non-linear) relationship between BMI and risk for COVID-19 related hospitalization, ICU admission, and death [28]. In our study, subjects with a BMI $\geq$ 30 kg/m$^2$ presented higher fatality with an odds ratio of 4.32, compared to subjects of a 23 $\leq$BMI < 25 kg/m$^2$. In addition, BMI < 18.5 kg/m$^2$ also showed an odds ratio of 3.97 (1.77–8.92) for fatal illness.

One recent remarkable study is the analysis of the UK biobank, even though preliminary data are available [29, 30]. They showed that BMI was strongly associated with positive results in the COVID-19 test and the risk of death related to COVID-19 [29]. Interestingly, both BMI and waist circumference were associated with testing positive for COVID-19 in a dose-response fashion [30]. However, another study using a two-sample multivariable Mendelian randomization failed to show the impact of body composition (waist circumference, trunk fat ratio) on COVID-19 susceptibility and severity; only BMI was significant [31]. Our study has a limitation in that it did not include body composition.

In infectious diseases, research on whether there are gender differences in the effect of BMI on mortality is limited. In a UK study [32], with the national mortality data of 3.6 million adults, the mortality specific outcome related to respiratory infection, the hazard ratio (HR) of underweight increased similarly to obesity, showing a U-shaped curve, which is consistent with our study results. Further, the UK study showed the difference between men and women in all-cause mortality among never-smokers. Underweight women were associated with a greater increase in HR than men, while obese men were associated with a greater increase in HR than women ($P_{\text{interaction}}$ < 0.0001). In other words, underweight and obesity were associated with an increase in the HR of all-cause mortality in both men and women, but the increased risk was greater in underweight women and obese men. This is consistent with our results shown in Fig 2. However, there are some differences between the study populations and the target outcomes, since our study aimed at evaluating the association between the fatality of COVID-19 and BMI. Additionally, we did not have information regarding smoking history. Furthermore, the sample size in our study was relatively small, and when divided by subgroup analysis, a similar pattern was shown as the UK study between men and women, although the $P_{\text{interaction}}$ did not show a significant value. Therefore, it is necessary to verify this through large-scale research. A Chinese study of 383 patients found that obesity, especially in men, significantly increased the risk of developing severe COVID-19 [9]. In this study, we found that there was a slight difference in the odds ratio and increasing pattern in those with BMI of < 18.5 and $\geq$ 25.0 kg/m$^2$ according to sex. Male sex was associated with an increased odds ratio of fatal illness (2.92 [1.9–4.48]) than the female sex. However, there was no definite evidence that men had a higher risk of severe COVID-19 due to BMI than women.

All-cause mortality due to obesity is attenuated by aging. Large-scale epidemiological studies have reported that heathy BMI shifts to the right in old age compared to young age [3, 32,

33]. Among the those with COVID-19, young obese patients ($< 60$ years old and BMI $> 35$ kg/m$^2$) were reported to be 3.6 times more likely to be admitted to the critical care unit than those with BMI $< 30$ kg/m$^2$ [34]. However, in our study, the incidence of severe COVID-19 in patients aged $< 60$ years was low (fatal illness was only 8 cases and critical illness was as small as 15 cases), so there was a limit to whether obesity was related to critical illness at a young age.

Obesity is regarded as an important public health issue, mainly related to chronic diseases. Obesity potentiates multiple cardiovascular risk factors and causes atherosclerosis. It also causes insulin resistance and reduces beta cell function [35]. It can cause functional immuno-logical deficits through dysregulation of the immune system, resulting in diseases mainly related to cardiometabolic problems and chronic inflammation [2]. However, in the recent SARS-CoV2 pandemic, obesity was reported to be associated with severe COVID-19, and its clinical significance in acute infections may be receiving new attention from researchers. In addition to chronic metabolic conditions, proposed mechanisms for the detrimental effect of obesity in severe COVID-19 are related to the cardiorespiratory system, increased thrombo-genicity, and immune hyper-reactivity [7, 36, 37]. Beyond these proposed mechanisms, COVID-19 can induce direct target organ damage, regardless of the underlying disease. Direct invasion or inflammation of cardiomyocyte is related to poor disease outcomes [18, 38, 39]. In the case of SARS, induced by SARS-CoV, the possibility of disruption of pancreatic beta cell function through pancreatic invasion through binding to the cellular entry, ACE-2, was con-firmed in an autopsy report [40]. However, there is paucity of evidence in COVID-19 pancrea-titis through direct beta cell involvement of SARS-CoV-2 [41]. Furthermore, there is no available data regarding whether obesity or underweight can augment the virus infection related to direct target organ damage by altering the susceptibility or ACE-2.

The present study has some limitations. Despite the advantage of nationwide data com-pared to previous studies related to COVID-19 and BMI, this study was limited to Koreans and did not include a diverse population. Although the fatality of BMI and COVID-19 was evaluated by adjusting for sex, age, and comorbid conditions, there was an issue in subgroup analysis because the sample was insufficient in terms of specific diseases. Furthermore, detailed information about comorbidities was not provided. The number of patients with BMI $< 18.5$, or $\geq 30$ kg/m$^2$, was small; 191 (4.6%) or 193 (4.7%), respectively. Therefore, there is a chance of being statistically underpowered or having an increased margin of error. Furthermore, we could not check the distribution of continuous BMI according to fatal illness status. The KCDA provided BMI data in a categorized form. Finally, it was an observational study and could not clearly explain causal relationships. Although the characteristics of acute infectious disease act as a limitation in the study of COVID-19, a large-scale prospective cohort study of various populations is of utmost important.

## Conclusions

We analyzed the relationship between BMI and criticality/fatality from COVID-19. In particu-lar, those with a BMI of $< 18.5$ kg/m$^2$ and $\geq 25.0$ kg/m$^2$ were found to have a higher associa-tion with critical and fatal illness compared to those with a BMI of 23.0–24.9 kg/m$^2$. Maintaining a healthy weight is important not only to prevent chronic cardiometabolic dis-eases, but also to improve the outcome of COVID-19.

## Supporting information

**S1 Table. Baseline characteristics of the study population according to disease severity.** (DOCX)

**S1 Fig. Number of critical and fatal illnesses, and fatalities according to body mass index.** The left y-axis presents bar graphs showing the number of critical and fatal illnesses, and the right y-axis presents as a line graph showing fatality.
(TIF)

## Acknowledgments

We acknowledge all health-care workers involved in the diagnosis and treatment of patients with COVID-19 in South Korea. We thank the Korea Disease Control & Prevention Agency, National Medical Center and the Health Information Manager in hospitals for their efforts in collecting the medical records.

## Author Contributions

**Conceptualization:** In Sook Kang, Kyoung Ae Kong.

**Data curation:** Kyoung Ae Kong.

**Formal analysis:** In Sook Kang, Kyoung Ae Kong.

**Investigation:** In Sook Kang, Kyoung Ae Kong.

**Methodology:** In Sook Kang.

**Supervision:** In Sook Kang.

**Validation:** In Sook Kang, Kyoung Ae Kong.

**Visualization:** In Sook Kang.

**Writing – original draft:** In Sook Kang.

**Writing – review & editing:** In Sook Kang, Kyoung Ae Kong.

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
