## [Decision Letter · Decision Letter 0]

12 Mar 2021

PONE-D-21-03568

Body mass index and fatality from coronavirus disease 2019: A nationwide epidemiological study in Korea

PLOS ONE

Dear Dr. Kang,

Thank you for submitting your manuscript to PLOS ONE. After careful consideration, we feel that it has merit but does not fully meet PLOS ONE’s publication criteria as it currently stands. Therefore, we invite you to submit a revised version of the manuscript that addresses the points raised during the review process.

We look forward to receiving your revised manuscript.

Kind regards,

Jie V Zhao

Academic Editor

PLOS ONE

Journal Requirements:

Reviewers' comments:

Reviewer's Responses to Questions

**Comments to the Author**

1. Is the manuscript technically sound, and do the data support the conclusions?

Reviewer #1: Yes

Reviewer #2: Partly

2. Has the statistical analysis been performed appropriately and rigorously? 

Reviewer #1: Yes

Reviewer #2: Yes

3. Have the authors made all data underlying the findings in their manuscript fully available?

Reviewer #1: No

Reviewer #2: No

4. Is the manuscript presented in an intelligible fashion and written in standard English?

Reviewer #1: Yes

Reviewer #2: No

5. Review Comments to the Author

Reviewer #1: This study investigated the relationship between BMI and COVID-19 fatal illness in a Korean sample. While this study is of quality, it can be improved in the following parts.

1. The introduction cited some studies about the relationship between BMI and severe COVID-19 illness. However, these studies are observational studies only and cannot provide high level evidence about the causation. There are several Mendelian randomisation studies on this relationship, such as Ponsford et al Circulation 2020 & Li et al BMC Medical Genomics 2021. These studies should be cited to give a more comprehensive review.

2. BMI is a continuous variable, but this study interestingly analysed it as a categorical variable. Transforming continuous variable to categorical would lose information. The study should provide the results for BMI as a continuous variable as well, and more importantly, to investigate if the U-shape still exists after using continuous BMI.

3. The study found a U-shape, which is interesting, especially for the underweight group. Table 1 shows that young people have the highest proportion of underweight people; however, young people have a low risk of fatal illness. So the results are a ‘paradox’. It’s good to report and check the distribution of continuous BMI by fatal illness status.

4. The results for the underweight group could be driven by the small size of this group (so the confidence interval is very wide). Conclusion based on this small number could be misleading. The authors should acknowledge this limitation and be cautious about their conclusion.

5. The authors interestingly used overweight group, rather than normal weight group, as the reference in their logistic regression. They justified this by stating that the overweight group had the lowest fatality rate. However, this is not a good justification, and the resulted ORs have little practical usage (because they are not compared with normal weight). Weight group should be the reference out of nature and intuition. The study should use the normal weight group as reference and report new results.

Reviewer #2: Please see below for my general comments on the Manuscript: Body mass index and fatality from coronavirus disease 2019: A nationwide epidemiological study in Korea. I have provided detailed feedback and questions regarding the methods, results, and interpretation of this analysis - both general comments/questions as well as line-specific. If my suggested analytical updates are made and thorough copy editing is completed, I believe this could add to the body of evidence around BMI and COVID-19 severity and fatality.

General comments/questions:

1. The manuscript has poor English grammar, and requires copy-editing for greater clarity.

2. The authors should doing a sensitivity analysis with the global BMI cut-points for underweight, normal weight, overweight and obesity. It would be interesting to see if there is a difference, given these Asia-Pacific population cut-points are relatively new and less widely used (which they discuss in the discussion but don't analyze)

3. The authors should provide definitions of each of the co-morbid conditions (self-report, taking medications), particularly for cancer – which cancers did they include?

Statistical Analysis: general comments/questions:

1.How did the authors account for the correlation between the various risk factors as well as potential mediating affects i.e. BMI as a mediator of diabetes)?

2.Why did the authors decide to categorize rather than use continuous variables (i.e. BMI, age as continuous variables)? Such multivariable analyses using continuous variables would add to the literature which typically categorizes BMI.

3.The authors report missing data in the Results (Table 1) but don’t explain how they handle missing data in the methods. Please clarify there.

4.Typically, the internal reference group for a cohort study should be the group with the least exposure (i.e. normal weight). While there is sufficient sample size to use the overweight as the reference, I am worried that using overweight instead of normal weight as the internal reference group makes the results confusing to interpret both within the study and in external comparison to other studies that use normal weight as reference. The OR for critical and fatal illness are both non-significant may be interpreted as a higher risk for critical and fatal illness unless very clearly discussed in the abstract, results and discussion. This could paint an impression that overweight is protective, when this may not be the case. For these two reasons, I’d suggest changing the reference group to normal weight. The authors provide some discussion on overweight BMI as a healthy in older adults. However, they do not do any further analysis to test this hypothesis. If they want to keep overweight as the reference category, an analysis of BMI - COVID19 severity/fatality by age (i.e. effect modification by age), would be helpful to determine if there is indeed this age effect.

Results general comments/questions:

1. At present the tables and figures do not provide sufficient information/footnotes to stand on their own for interpretation. For instance, in Table 2, the footnote needs to indicate that age is for every 10 years (categorical instead of continuous) and provide details on how critical and fatal illness are defined, how co-morbid conditions are defined, and what covariates were used in analysis

2. Figure 2: It is surprising to me that these two have the exact same p-interaction value (for female/male and critical illness, fatal illness) to the 0.0001 confidence level. I’d advise the authors double check this and confirm, especially considering how different the trends/figures look.

3. Include 95% CI in the narrative text in addition to the tables and figures

4. Improve interpretation sentences for odds ratios.

Discussion general comments/questions:

1. The authors include new results from the sub-group analyses in their discussion instead of summarizing and interpreting findings from their results.

2. The authors focus their results on the literature around BMI and all-cause mortality. This is somewhat related, but it would bet better to review literature more specific to BMI and infection-related fatality.

3. No mention of alternative exposure measures for overweight/obesity like waist circumference or waist to hip ratio. Even if not possible with this data set, it would be good to mention limitations of BMI as an exposure measure.

4. Need more discussion on small sample size for underweight and obese II and how miss affected the certainty in the odds ratio estimates.

I suggest the authors add into their discussion and intro this new U.S. BMI and COVID-19 severity and death, which found a similar non-linear relationship between BMI and COVID-19 severity/death (increased risk at underweight and obese level BMIs)

Kompaniyets L, Goodman AB, Belay B, et al. Body Mass Index and Risk for COVID-19–Related Hospitalization, Intensive Care Unit Admission, Invasive Mechanical Ventilation, and Death — United States, March–December 2020. MMWR Morb Mortal Wkly Rep. ePub: 8 March 2021. DOI: http://dx.doi.org/10.15585/mmwr.mm7010e4external icon.

Line specific Comments:

Abstract:

32- 34: No mention of OR for normal weight in the abstract.

36-37: Conclusion explains underweight and obese have increased risk for critical/fatal illness than normal and overweight, but OR is only for overweight as reference.

Introduction:

45 : poor grammar – “as the increasing period of social distancing”, “more like to eat unhealthy food”

48-50: Its unclear why the author is mentioning U-shaped curve for BMI and all cause, CVD mortality, when the outcome here is COVID-19 related severe illness and fatality. Better to remove and cite literature on the association between BMI and flu-like virus severe illness and fatality (which you do a bit later).

51-57: There is now far more global data that can be cited regarding cohorts of COVID-19 severity and hospitalization with BMI as a risk factor. Consider citing others (particularly within Asia region for comparability)

58: clarify if cardiac disease means Cardiovascular disease (CVD)?

65-66 : bad English grammar

68: “associated with COVID-19…” and “obesity as a risk factor for COVID-19” Be more specific. Are you referring to COVID-19 infection or COVID-19 severe illness/fatality? I don’t know of any evidence suggesting obese have a higher risk for being INFECTED, though there is substantial data for SEVERE ILLNESS/FATALITY?

74-75: this line about Korean vs. Westerns having lower obesity and higher underweight prevalence belongs earlier in the introduction. Move and add more about the prevalence (cite national level stats for Korea)

Data Source and Study Population

83: “almost all confirmed cases” – what does this mean? Can you provide an estimate of the percentage of tracking currently occurring?

88-91: how has the exclusion of 1487 patients affected representativeness of the data? In particular, are those individuals without available BMI values systematically different than those with BM data? This could lead to serious selection bias. Include tables/data on representativeness of selected data vs. missing data.

Statistical Analysis

118: incorrect use of the term multivariate- this is a multivariable** logistic regression. Multivariate refresh to modeling of data that are often derived from longitudinal studies (repeated measures)m which this is not.

118: should be COVID-19 critical illness and COVID-19 fatal illness

127- 131: language around the stratified analyses is unclear at present. It sounds like the authors did an overall multivariable logistic regression, controlling for all potential confounders and then did stratified analyses by sex and each co-morbid condition separately. Add in language around stratified analysis for greater clarity.

129-131: this last statement is also confusing. It is unclear from what the authors are saying whether, when stratifying by each comorbidity, they are calculating the odds ratios only for patients who do vs. do NOT have the target disease?

Table 1

- Include percentage of total population for each BMI (i.e. 1741 is 42% of total patients) as you have written this in your narrative results.

- Is the p-value for the statistically significant differences within a given covariate (age group, sex, etc.) across BMI categories? Or is it a p-value for statistically significant differences within a BMI category across covariates (across age groups, sexes, etc.). For the results text it seems to be the former, but please clarify in methods and in table footnotes what the p-value refers to

161: start new paragraph when start talking about the association/odds ratio results. “Fatal illness was significantly related to the combined comorbidity” – unclear what this means as written.

175 -176: unclear what the authors mean by “did not change the aspect of underweight”. This is not common statistical language.

Figure 2: It is surprising to me that these two have the exact same p-interaction value (for female/male and critical illness, fatal illness) to the 0.0001 confidence level. I’d advise the authors double check this and confirm, especially considering how different the trends/figures look.

Discussion:

203-209: This review of sub-group analyses should be moved to the results section. The authors should only include a short summary of this analysis and its implications in the discussion section.

223-224: It is not clear what the authors mean by “contradictory results were reported in obesity according to other studies”. I suggest they make this clear.

225-244: Remove or reduce the detailed comparison to UK literature on BMI and overall mortality (non-specific to infection). A mention of the overall association between BMI and mortality is ok, but not a detail comparison as the authors here are specifically looking at COVID-19 severe illness and fatality. Better to compare to literature on similar viral infections or COVID19 studies.

245-255: same is true here. The discussion should be focused more on literature about the actual exposures and outcomes of interest in this paper, not a more general topic like all cause mortality.

256: “The BMI value for healthy weight varied with age” – is this a finding of the analysis or something gleaned from the literature. As written, it is confusing whether this is a general statement or a specific finding.

258-259: This is an interesting statement. Did the authors do any analysis to confirm whether there was effect modification by age for the OR in overweight vs. normal weight in the study. Would be necessary to confirm this with data.

266-268: poor English grammar, consider revising.

266-284: discussion around Asian specific BMI cut points – did the authors consider doing a sensitivity analysis with the general WHO BMI cut-points? This would be a helpful comparison to see how much these differential BMI cut-points made a difference in the reported odds ratios.

279: typo – should be 25.0-29.9 kg/m2, not 2.5-29.9 kg/m2

286-287: mention of how obesity potentiates multiple cardiovascular risk factors, but they did not discuss how they addressed this in their methods. See general statistical analysis question above

315: the authors looked at the relationships between BMI and critical and fatal illness from COVID-19 (not just fatality from COVID-19). Revise how this is written.

6. PLOS authors have the option to publish the peer review history of their article (what does this mean?). If published, this will include your full peer review and any attached files.

Reviewer #1: No

Reviewer #2: No

---

## [Author Response · Author response to Decision Letter 0]

26 May 2021

We deeply appreciate the Reviewer for his/her time and input. We have carefully considered the reviewer’s comments and have addressed them point-by-point.

We attached this as a file.

---

## [Decision Letter · Decision Letter 1]

10 Jun 2021

Body mass index and severity/fatality from coronavirus disease 2019: A nationwide epidemiological study in Korea

PONE-D-21-03568R1

Dear Dr. Kang,

We’re pleased to inform you that your manuscript has been judged scientifically suitable for publication and will be formally accepted for publication once it meets all outstanding technical requirements.

Kind regards,

Jie V Zhao

Academic Editor

PLOS ONE

Additional Editor Comments (optional):

Reviewers' comments:

Reviewer's Responses to Questions

**Comments to the Author**

1. If the authors have adequately addressed your comments raised in a previous round of review and you feel that this manuscript is now acceptable for publication, you may indicate that here to bypass the “Comments to the Author” section, enter your conflict of interest statement in the “Confidential to Editor” section, and submit your "Accept" recommendation.

Reviewer #1: All comments have been addressed

2. Is the manuscript technically sound, and do the data support the conclusions?

Reviewer #1: (No Response)

3. Has the statistical analysis been performed appropriately and rigorously? 

Reviewer #1: (No Response)

4. Have the authors made all data underlying the findings in their manuscript fully available?

Reviewer #1: (No Response)

5. Is the manuscript presented in an intelligible fashion and written in standard English?

Reviewer #1: (No Response)

6. Review Comments to the Author

Reviewer #1: (No Response)

7. PLOS authors have the option to publish the peer review history of their article (what does this mean?). If published, this will include your full peer review and any attached files.

Reviewer #1: No

---

## [Editor Report · Acceptance letter]

14 Jun 2021

PONE-D-21-03568R1 

Body mass index and severity/fatality from coronavirus disease 2019: A nationwide epidemiological study in Korea 

Dear Dr. Kang:

I'm pleased to inform you that your manuscript has been deemed suitable for publication in PLOS ONE. Congratulations! Your manuscript is now with our production department. 

Kind regards, 

on behalf of

Dr. Jie V Zhao 

Academic Editor

PLOS ONE